# Enhancing Argument Structure Extraction with Efficient Leverage of Contextual Information

**Yun Luo** [1], **Zhen Yang** [2], **Fandong Meng** [2], **Yingjie Li** [1], **Jie Zhou** [2], **Yue Zhang** [1,3] *

[1] School of Engineering, Westlake University, Hangzhou, China.

[2] Pattern Recognition Center, WeChat AI, Tencent Inc, Beijing, China.

[3] Institute of Advanced Technology, Westlake Institute for Advanced Study, Hangzhou, China.

{luoyun, liyingjie,zhangyue}@westlake.edu.cn

{zieenyang, fandongmeng, withtomzhou}@tentent.com

## Abstract

Argument structure extraction (ASE) aims to identify the discourse structure of arguments within documents. Previous research has demonstrated that contextual information is crucial for developing an effective ASE model. However, we observe that merely concatenating sentences in a contextual window does not fully utilize contextual information and can sometimes lead to excessive attention on less informative sentences. To tackle this challenge, we propose an Efficient Context-aware ASE model (ECASE) that fully exploits contextual information by enhancing modeling capacity and augmenting training data. Specifically, we introduce a sequence-attention module and distance-weighted similarity loss to aggregate contextual information and argumentative information. Additionally, we augment the training data by randomly masking discourse markers and sentences, which reduces the model's reliance on specific words or less informative sentences. Our experiments on five datasets from various domains demonstrate that our model achieves state-of-the-art performance. Furthermore, ablation studies confirm the effectiveness of each module in our model.

## 1 Introduction

Argument structure extraction (ASE) aims to identify the argumentative discourse structure in documents (Peldszus and Stede, 2013; Cabrio and Villata, 2018; Lawrence and Reed, 2019; Li et al., 2020). As a basis of evidence-based reasoning applications, ASE has attracted increasing attention from researchers in a wide spectrum of domains, such as legal documents (Palau and Moens, 2009; Lippi and Torroni, 2016; Poudyal et al., 2020), online posts (Cardie et al., 2008; Boltužić and Šnajder, 2014; Park and Cardie, 2018; Hua and Wang, 2017), and scientific articles (Mayer et al., 2020; Al Khatib et al., 2021). [1] Figure 1 shows an exam-

---

[1]For details about the related work, we refer the readers to Appendix A.

**Peer Review**

(1) It is clear that the problem studied in this paper is interesting.

(2) **However**, after reading through the manuscript, it is not clear to me what are the real contributions made in this paper.

(3) I also failed to find any rigorous results on generalization bounds.

(4) **Therefore**, I cannot recommend the acceptance of this paper.

*Support*

Figure 1: An example of argument structure extraction in peer reviews. The argumentative discourse markers are marked in red.

ple where the second and third sentences describe the shortcomings of the reviewed paper and support the fourth conclusion sentence.

It has been verified that the contextual information of documents plays a crucial role in detecting argumentative relations (Nguyen and Litman, 2016; Opitz and Frank, 2019; Hua and Wang, 2022). According to our preliminary experiments based on the Context-aware ASE model (CASE) (Hua and Wang, 2022), several findings motivate us to further explore the efficient use of contextual information. Firstly, based on our experimental results shown in Figure 2(a), we find that the performance of the model is initially improved with the increase of the context window, but then decreases as the context window continues to grow. These results indicate that contextual information can enhance the performance of the model, but the excessively long context may cause the model to overly focus on some irrelevant sentence information and behave relatively weak performance. Secondly, the performance of CASE decreases as the distances of argumentative pairs increase, as shown in Figure 2(b), which suggests that detecting argumentative relations over long distances is still challenging for CASE. It also implies that there is still room for improvement by making full use of contextual information rather than simply concatenating the sentences in a context window. Thirdly, Figure

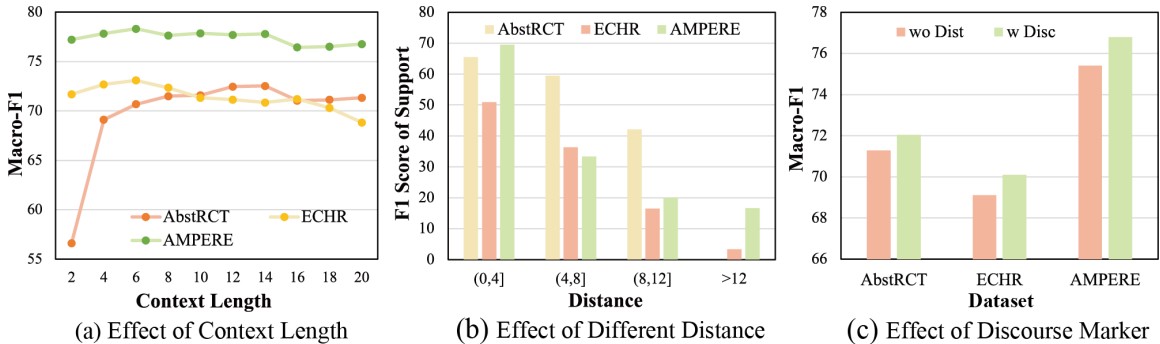

Figure 2: Analysis of the CASE model in three different datasets (AbstRCT, ECHR, and AMPERE). (a) shows the performance when the context length varies; (b) shows the macro-F1 scores with respect to the distance between argumentative pairs; (c) refers to the model performance on the samples with/without discourse markers.

2(c) shows that the model mostly achieves better performance in argumentative pairs where there are discourse markers, such as 'so', 'thus', and et. These specific words are regarded as significant signals for identifying argumentative relations (Stab and Gurevych, 2014; Hua and Wang, 2022), but they may hinder the model from fully exploiting the contextual information between sentences.

Based on the above observations, we propose an Efficient Context-aware ASE model (ECASE), which enhances the leverage of contextual information from two perspectives, i.e., modeling capacity and data augmentation. Regarding the modeling capacity, we adopt a sequence attention module on RoBERTa (Liu et al., 2019) to aggregate the contextual information and optimize the attention distribution between sentences. We also use a distance-weighted similarity loss to reinforce the representation similarity in view of argumentative relations. In terms of data augmentation, we randomly mask the discourse markers and sentences in training sets to mitigate the model's dependence on specific words and less informative sentences. Hence, the model is encouraged to thoroughly comprehend the contextual information.

Experiments on five datasets from different domains show that our model achieves state-of-the-art performance in both the head-given setting and the end-to-end setting of the ASE task. Ablation studies also demonstrate the effectiveness of each module in our model. The codes are released in https://github.com/LuoXiaoHeics/ECASE.

## 2 Method

### 2.1 Task Formulation

We formulate the ASE task as a classification task. Formally, let $D = \{d^i\}_{i=1}^N$ be a dataset with $N$ documents, each consisting of several propositions $\{s_k\}^i$. The task is to identify the existence of $attack$, $support$ link from $s_j$ to $s_k$, or $no-rel$ (no relation) between $s_j$ and $s_k$. We consider the task both in an end-to-end setting (considering all proposition pairs) and a head-given setting where the head propositions are given in advance. The framework of our model is illustrated in Figure 3.

### 2.2 Modeling Capacity Enhancement

#### 2.2.1 Encoding with Sentence-level Attention

Following Hua and Wang (2022), we build our model on top of RoBERTa (Liu et al., 2019), which is a widely-used encoder-only pre-trained language model. Given a head proposition $s_j$, we concatenate it with surrounding sentences, which provide the contextual information, including $L$ propositions before and after $s_j$. Other propositions in the context are regarded as tail propositions. $[CLS]$ tokens are added to separate the propositions and serve as their representations. The representations of head proposition $s_j$ or other tail propositions can be obtained as follows:

$$H_j^w = RoBERTa(Context)[CLS]_j. \quad (1)$$

To further extract the relation between sentences, we use a sentence-level attention module to aggregate the contextual information. Specifically, we concatenate the sentence representations in the contextual window and feed them into a multi-head self-attention module (Vaswani et al., 2017) to aggregate the contextual information, which can be formulated as follows:

$$H_j^s = SeqAtten([H_j^w;...;H_L^w])[j], \quad (2)$$

where each row of query, key, and value in the self-attention module corresponds to a sentence

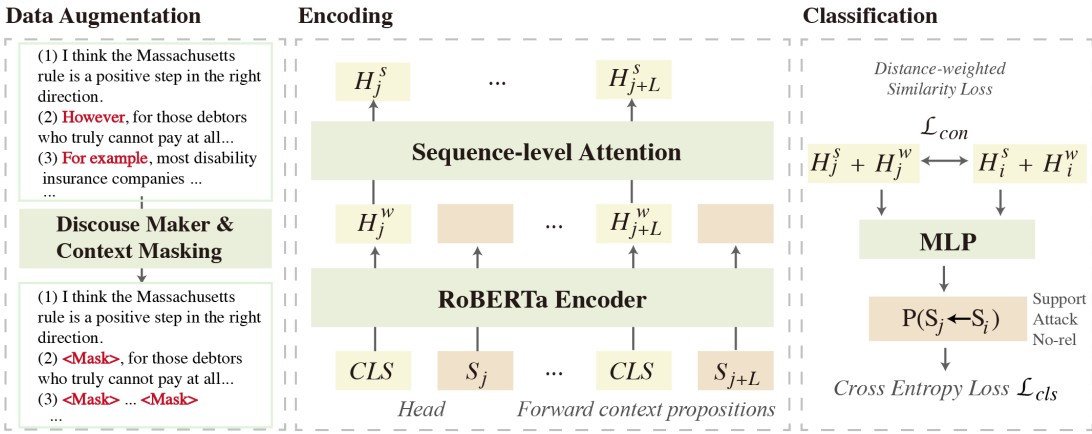

Figure 3: The model framework of our proposed argument structure extraction model.

representation. The representations from the token level and sentence level are mixed in a simple way of addition: $H_j = H_j^w + H_j^s$.

Then, we utilize a multi-layer perceptron for calculating the label distribution of the head-tail pairs (such as $(j, k)$), which can be shown as:

$$P(y|s_j, s_k) = Softmax(tanh([H_j; H_k] \cdot \mathbf{W}_1) \cdot \mathbf{W}_2) \quad (3)$$

where $\mathbf{W}_1$, $\mathbf{W}_2$ are trainable parameters.

### 2.2.2 Argumentative Relation Regularization

We further propose a similarity loss to enhance the semantic information of the sentence embeddings (Reimers and Gurevych, 2019; Wieting and Gimpel, 2018; Luo et al., 2022a), which is formulated as:

$$\mathcal{L}_{con} = \begin{cases} 1 + exp(-\frac{d_{ij}}{d})sim(H_i, H_j) & \text{if } y = 0, \\ 1 - exp(\frac{d_{ij}}{d} - 1)sim(H_i, H_j) & \text{if } y \neq 0. \end{cases} \quad (4)$$

Specifically, the loss aims to pull head-tail representations with argumentative relations ($y \neq 0$) and push away them with *no-rel* links ($y = 0$) through optimizing cosine similarity $sim(\cdot, \cdot)$. Considering that the representations encoded by PLMs have a relatively weak similarity over long distances and vice versa, we use a distance weight to reinforce the similarity differences. The weight is calculated by the exponent of the normalized distance, where $d_{ij}$ is the distance between representations of sentence $s_i$ and $s_j$ and $d$ is the maximum token number.

### 2.3 Data Augmentation

In this paper, we propose both word-level and sentence-level data augmentation for the ASE task. First, our observations indicate that models tend to perform better on the head-tail pairs with discourse markers. Thus, in word-level data augmentation,

| | AMPERE | Essays | AbstRCT | ECHR | CDCP |
|---|---|---|---|---|---|
| # Prop. | 10,386 | 12,373 | 5,693 | 6,331 | 4,932 |
| # Supp. | 3,370 | 3,613 | 2,402 | 1,946 | 1,426 |
| # Att. | 266 | 219 | 70 | 0 | 0 |
| # Head | 2,268 | 1,707 | 1,138 | 741 | 1,037 |
| % Disc | 14.58% | 10.44% | 13.94% | 14.08% | 12.30% |

Table 1: Dataset statistic. % Disc refers to the ratio of propositions with discourse makers.

we randomly mask discourse markers (with probability $p_w$) to encourage the model to learn more contextual information in sentences and rely less on discourse markers. 18 discourse markers are selected from the PDTB manual (Prasad et al., 2006) following Hua and Wang (2022) (Table 3).

Second, since the long context could introduce noise into the sequences, the attention to the less informative sentences might have a negative impact on identifying specific head-tail pairs. Therefore, we generate samples from original data by randomly (with probability $p_s$) masking some sentences to enhance the comprehension of the contextual information and mitigate the excessive attention on certain sentences.

### 2.4 Training

The overall training loss is calculated as follows: $\mathcal{L} = \mathcal{L}_{cls} + \lambda\mathcal{L}_{con}$ where $\lambda$ is the hyper-parameter to control the weight of representation similarity.

## 3 Experiments Setup

We carry out experiments on five ASE datasets from different domains, namely AMPERE, Essays, AbstRCT, ECHR, and CDCP, whose statistics are shown in Table 1 (details in Appendix B). Our model is compared with several baseline models: 1) two classic lexical ASE models **SVM-linear** and **SVM-RBF** (Stab and Gurevych, 2017); 2) **SEQ-**

| | AMPERE | | | Essays | | | AbstRCT | | ECHR | | CDCP | |
|---|---|---|---|---|---|---|---|---|---|---|---|---|
| | Supp. | Atk. | Macro | Supp. | Atk. | Macro | Supp. | Macro | Supp. | Macro | Supp. | Macro |
| SVM-Linear | - | - | 24.82 | - | - | 28.69 | - | - | - | 21.18 | - | 29.01 |
| SVM-RBF | - | - | 26.38 | - | - | 31.68 | - | | - | 21.36 | - | 30.34 |
| SEQPAIR | 17.34 | 7.40 | 26.42 | 31.42 | 24.32 | 30.37 | 17.32 | 32.34 | 23.11 | 33.23 | 14.16 | 28.44 |
| *Head Given* | | | | | | | | | | | | |
| SEQCON-10 | 43.04 | 49.67 | 63.34 | 47.71 | 30.14 | 57.40 | 60.60 | 69.20 | 41.81 | 65.48 | 38.34 | 63.10 |
| CASE-10 | 63.86 | 70.41 | 77.62 | 72.03 | 40.61 | 70.19 | 63.39 | 71.08 | 42.35 | 71.11 | 45.73 | 69.76 |
| CASE-20 | 64.45 | 68.56 | 77.41 | 72.14 | 40.14 | 69.13 | 63.35 | 70.94 | 35.18 | 69.35 | 45.63 | 69.56 |
| CASE-20* | - | - | 77.64 | - | - | 71.30 | - | - | - | 70.82 | - | 70.37 |
| ECASE-10 | **67.37** | **71.14** | **79.01** | **75.91** | **44.40** | **72.38** | 63.91 | **72.17** | **47.41** | **73.15** | 51.16 | 72.18 |
| ECASE-20 | 66.12 | 71.10 | 78.46 | 74.21 | 42.30 | 72.36 | **64.42** | 72.10 | 41.55 | 70.46 | **51.85** | **72.36** |
| *End-to-End* | | | | | | | | | | | | |
| CASE-20 | 60.33 | **60.55** | **73.23** | 70.73 | 36.31 | 68.58 | **64.85** | 72.79 | 31.77 | 65.66 | 39.56 | 66.22 |
| ECASE-20 | **60.82** | 57.56 | 72.82 | **73.74** | **37.54** | 69.51 | 63.58 | **73.59** | 39.63 | 69.49 | 46.87 | 70.16 |

Table 2: Results of our model on five different ASE datasets. '-' for not publishing, and '*' refers to the results reported in the original paper, considering pre-training on unlabeled data for domain adaptation. '-10' and '-20' refer to the context length. 'Supp.' and 'Atk.' represent the F1 scores of 'Support' and 'Attack', respectively.

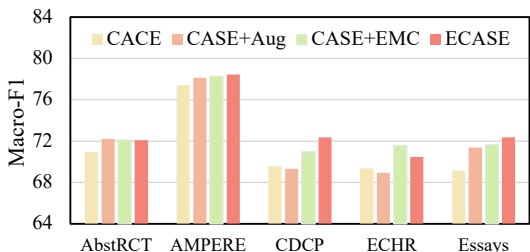

Figure 4: Ablation study of the models with context length 20 in the head-given setting. 'Aug' and 'EMC' refer to data augmentation and the modules for enhancing modeling capacity.

**PAIR** (Devlin et al., 2019) uses BERT to encode the head and tail in single sentences separately and concatenates their representations to predict the argumentative relation; 3) **SEQCON**, a context-aware version of SEQPAIR, where the head and tail are encoded in the context of each other, and $[CLS]$ representations of the head and tail are concatenated for classification; 4) **CASE** (Hua and Wang, 2022) concatenates the head sentence with forward context propositions and backward context propositions in a window for encoding. In the experiments of ECASE and reproduced CASE, we keep the same training hyper-parameters such as training steps (Appendix C).

## 4 Results

### 4.1 Overall Results

The main results are shown in Table 2. Specifically, in the head-given setting, ECASE-10 achieves state-of-the-art performance (79.01%, 77.38%, 72.17%, 73.15%, and 72.16% macro-F1 scores) compared with other baselines. When increasing the context length of ECASE to 20, the model also performs better compared with CASE-20. Although shorter contexts still perform better in most datasets

for ECASE-10 compared with ECASE-20, the gap between them is relatively smaller compared with CASE. Notably, the model performance of ECASE-20 (72.36%) in CDCD is even higher than that of ECASE-10 (72.18%), indicating that our model can efficiently extract contextual information while mitigating excessive attention to less informative sentences to some degree. Moreover, models with backward and forward context inputs, such as CASE and ECASE, outperform SEQPAIR and SEQCON-10, demonstrating the effectiveness of the input form and contextual information.

In the end-to-end setting, ECASE-20s achieve stronger performance compared with CASE-20 in most datasets. The results also demonstrate that the efficient use of contextual information can boost the identification of argumentative structure in raw discourses. However, the performance of ECASE-20s in the end-to-end setting is generally lower than in the head-given settings. This suggests that detecting argumentative relations becomes increasingly difficult in the end-to-end scenario due to the small ratio of *support* and *attack* labels. But the performance of ECASE-20 on AbstRCT is higher in such a setting which indicates in the dataset the *no-rel* samples are significant for models to distinguish the argumentative relations and the argumentative pairs are expressed more evidently compared with *no-rel* sentence pairs. [2]

### 4.2 Ablation Study

We also conduct ablation studies to highlight the effectiveness of each module in our model. Figure 4 presents the results, indicating that the inclusion of

---

[2] Further analysis of the context length, discourse makers, and case study of GPT3.5 can be found in Appendix D&E.

data augmentation (CASE+Aug) and the modules for enhancing modeling capacity (CASE+EMC) can enhance the performance of CASEs. We also observe that compared with data augmentation, the refinement of the modeling capacity brings a more critical improvement. ECASEs achieve the most significant performance in most datasets indicating the effectiveness of our model.

## 5 Conclusion

In this study, we proposed an efficient context-aware ASE model by enhancing modeling capacity and augmenting training data. Our experiments demonstrated the effectiveness of our model, and the ablation study proved the significance of each module. Our study highlighted the essential role of efficient use of contextual information.

## Limitations

According to preliminary experiments, we find it difficult for both CASE and ECASE to identify the $attack$ relations in AbstRCT dataset without using data for unsupervised training. The samples of $attack$ are extremely scarce in the dataset. Thus we regard AbstRCT as a two-label task, where the relation $attack$ is regarded as $no - rel$. And in this study, we only consider the encoder-only model RoBERTa as our backbone network, without the consideration of encoder-decoder or decoder-only models. In comparison with GPT-3.5. we do not test GPT3.5-turbo in full test data of ASE but show some randomly selected samples to demonstrate its performance because of the complex output form as well as the time and expense costs. But the results have shown the feature of the output from GPT3.5-turbo in ASE tasks.

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

## A  Related Work

Argument structure extraction has attracted increasing attention in recent years, which is a subtask of opinion mining (Lippi and Torroni, 2016; Cardie et al., 2008; Lawrence and Reed, 2019). The task aims to identify the structure or relations of the argument propositions. It differs from sentiment analysis (Liu, 2022) and stance detection (Luo et al., 2023, 2022b) since argument mining plays a role in expressing and promoting an opinion, i.e. determining why individuals hold the opinions but not what opinion they hold (Lawrence and Reed, 2019). It can be divided into two subtasks as premise detection and relation classification, where the former aims to identify the targeted propositions (head) and the latter requires classifying the relations of other propositions (tail) to the head. Research in different levels is proposed for analyzing the discourse structure such as segment level (Sun et al., 2022; Bao et al., 2022; Ye and Teufel, 2021) and proposition level (Hua and Wang, 2017, 2022).

Early methods model the task based on discourse parsing (Peldszus and Stede, 2013, 2015; Stab and Gurevych, 2017) and some methods use statistical and human-made features for classification (Stab and Gurevych, 2014; Niculae et al., 2017). In recent years, ASE models based on pre-trained language models are proposed and achieve great performance (Sun et al., 2022; Bao et al., 2022; Hua and Wang, 2022). For example, Sun et al. (2022) use BERT (Devlin et al., 2019) as the backbone network and use probing for further extraction of the semantic information in the language model. Hua and Wang (2017) propose a context-aware model based on RoBERTa (Liu et al., 2019) encoding the head propositions in a window for the ASE task. Our model is built on Hua and Wang (2022), and further considers how to efficiently leverage the contextual information.

## B  Datasets

Following Hua and Wang (2022), we adopt the datasets across the domains of peer review, essays, biomedical paper abstracts, online user discussions, and legal documents. The statistics of the datasets are shown in Table 3, and the details are described as follows:

**Peer Reviews.** The dataset AMPERE consists of 400 ICLR 2018 paper reviews collected from Open-

Review[3] (Hua and Wang, 2022). Each review in the dataset is annotated with segmented propositions and the corresponding types such as *evaluation, request, fact*, and so on. The relations of *support* of *attack* are further labeled among the propositions. We use 300 reviews for training, 20 for validation, and 80 for testing.

**Essays.** The dataset Essays contains 402 essays collected by Stab and Gurevych (2017) from [4]. The propositions are annotated in the sub-sentence level with corresponding types such as *premise, claim*, or *major claim*. The relations (support or attack) are annotated from a *premise* to a *claim* or to another *premise*. The dataset is split into 228, 40, and 80 for training, validation, and testing.

**Biomedical Paper Abstract.** The dataset AbstRCT contains 700 paper abstracts from PubMed (Mayer et al., 2020). Note that the dataset contains fewer propositions compared with the above ones, with only 70 attack links. For simplicity, we regard attack as *no-rel*, and only make classification on *support* and *no-rel* since the ratio of $attack$ is significantly low.

**Legal Documents.** The dataset ECHR (Poudyal et al., 2020) contains 42 documents from the European Court of Human Rights, which annotated the links from premises to conclusions.

**Online User Comments.** Park and Cardie (2018) (CDCP) contains annotated comments related to Consumer Debt Collection Practices, where labels of supporting relations are given.

## C  Implementation Details

We perform experiments using the official pre-trained RoBERTa (Liu et al., 2019) (roberta-base) provided by Huggingface [5]. The models are trained on 1 GPU (Tesla V100) using the Adam optimizer (Kingma and Ba, 2014). The learning rate is 1e-5, and the scheduler is set constant. We train our model in 15 epochs. We hyper-tune ECASE with $\lambda \in \{0.1, 0.05, 0.01, 0.001\}$, and find that the experimental results are relatively stable, indicating that the model is less sensitive, and we take $\lambda = 0.01$ for our experiments. The probability of $p_s$ and $p_w$ are 0.15 and 0.5, respectively. We evaluate the model performance based on macro-F1 scores. The reported results are averaged with 5 different random seeds. The reproduced exper-

---

[3]https://openreview.net/
[4]http://essaysforum.com
[5]https://huggingface.co/

| because | therefore | however | although |
|---|---|---|---|
| though | nevertheless | nonetheless | thus |
| hence | consequently | for this reason | due to |
| in particular | particularly | specifically | in fact |
| actually | but | | |

Table 3: Discourse markers of PDTB manual.

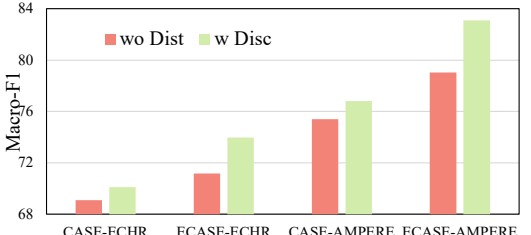

Figure 5: Results of CASE and ECASE w/o the discourse markers.

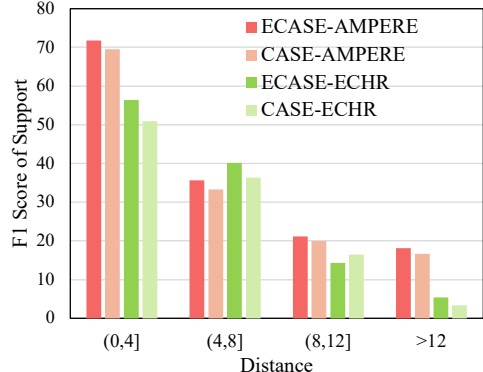

Figure 6: Results of CASE and ECASE with different distances of the head-tail pairs.

iments of CASE are based on the released codes and follow the same setting as our model.

## D    Case Study with GPT3.5

Recently, LLMs in generative schema become dominant in the area of natural language processing such as ChatGPT [6] and Claude [7]. But it has been verified that in-context learning based on LLMs performs weaker compared with fine-tuned small language models, especially in tasks with complex outputs (Ma et al., 2023). Here we also show some cases tested on GPT3.5-turbo in the head-given setting (Table 4). Note that in the zero-shot setting, the performance of GPT3.5 is not excellent in solving the task of ASE and the results indicate that developing small models with supervised data for the specific task of ASE is still significant and valuable.

The instruction fed into GPT is that 'Given the text {document}, what is the argumentative relation of other sentences to the sentence {head}, support, against or no relation? And show which sentence.' We observe that the model tends to give a consistent prediction of tail sentences to head sentences. For example, in case 3, GPT3.5 replies as: 'The first, second, and third sentences support the argument made in the fourth sentence. They state that the problem studied in the paper is interesting but that the manuscript does not make clear contributions or provide rigorous results on generalization bounds. The fourth sentence concludes that, based on these shortcomings, the paper cannot be recommended for acceptance.' But the first sentence obviously

does not support the conclusion.

Meanwhile, in some situations with complex logic, both GPT3.5 and ECASE can not give correct answers. For example, in case 5 GPT3.5 replies as: 'The argumentative relation of other sentences to the first sentence is attack. The second sentence is attacking the first sentence by saying that texts do work for consumers, but they have to pay for it. The third sentence further attacks the first sentence by stating that charging consumers for texts is against the law and rightly so.'

## E    Further Analysis

Here we also compare our model with CASE corresponding to the experiments in the introduction (Figures 5 and 6). As we can observe, our model can boost the performance both in the pairs with or without discourse marker, and the sentence pairs with different distances of head-tail pairs. The results demonstrate that our model makes more efficient use of contextual information and achieves stronger performance in ASE tasks.

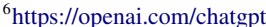

[6]https://openai.com/chatgpt
[7]https://www.anthropic.com/index/introducing-claude

| Document | Relation. | GPT3.5 | ECASE |
|---|---|---|---|
| The paper studies the problem of DNN loss function design for reducing intra-class variance in the output feature space. The key contribution is proposing an isotropic variant of the softmax loss that can balance the accuracy of classification and compactness of individual class. The proposed loss has been compared extensively against a number of closely related approaches in methodology. Numerical results on benchmark datasets show some improvement of the proposed loss over softmax loss and center loss (Wen et al., 2016), when applied to distance-based classifiers such as k-NN and k-means. Pros: - The idea of isotropic normalization for enhancing compactness of class is well motivated. The paper is mostly clearly organized and presented. Numerical study shows some promise of the proposed method. Cons: - The novelty of method is mostly incremental given the prior work of (Wen et al., 2016) which has provided a different isotropic variant of softmax loss. | (7,4,Supp) | no-rel | (7,4,Supp) |
| The paper is not anonymized. In page 2, the first line, the authors revealed [15] is a self-citation. And [15] is not anonumized in the reference list. | (2,1,Supp) (3,1,Supp) | (2,1,Supp) (3,1,Supp) | (2,1,Supp) (3,1,Supp) |
| It is clear that the problem studied in this paper is interesting. However, after reading through the manuscript, it is not clear to me what are the real contributions made in this paper. I also failed to find any rigorous results on generalization bounds. In this way, I cannot recommend the acceptance of this paper. | (4,2,Supp) (4,3,Supp) | (4,1,Supp) (4,2,Supp) (4,3,Supp) | (4,2,Supp) (4,3,Supp) |
| Why should the consumer pay a filing fee at all if the collector is at fault? That could be a hardship on many people. The collection agencies need to follow the rules of doing their validation correctly. This would not be an issue. | (1,2,Supp) | (1,3,Supp) | (1,2,Supp) |
| Texts will not work for consumers. Consumers must pay for texts. And this is already against the law will rightly so. | (1,2,Supp) (1,3,Supp) | (1,2,Atk) (1,3,Atk) | no-rel |
| I would like to have some protection from the calls I have received over several years from debt collection services looking for a woman who does not live at this address and has never lived at this address. I keep getting reassurances that my number will be removed, but the calls continue. | (1,2,Supp) | (1,2,Supp) | no-rel |

Table 4: Case Study of ASE. In the column of Relation., the first term is head, the second term is tail, and the third term is the argumentative relation from tails to heads. The cases are tested in the head-given setting.