# OpenReview forum: "Enhancing Argument Structure Extraction with Efficient Leverage of Contextual Information"
_EMNLP/2023/Conference — EMNLP 2023 Findings_

### Official Review · Reviewer_xDHq · 2023-08-01

**Soundness:** 2

**Excitement:**

3: Ambivalent: It has merits (e.g., it reports state-of-the-art results, the idea is nice), but there are key weaknesses (e.g., it describes incremental work), and it can significantly benefit from another round of revision. However, I won't object to accepting it if my co-reviewers champion it.

**Paper Topic And Main Contributions:**

The topic of this paper is about approaches for data- and compute efficiency.
The main contributions of this paper are as follows:
(1)This paper proposes an Efficient Context-aware ASE model (ECASE) that fully exploits contextual information. Specifically, introduce a sequence-attention module, a distance-weighted similarity loss, and a discourse marker.
(2)The comparative experiments on five datasets from various domains demonstrate that the proposed model achieves state-of-the-art performance.


**Questions For The Authors:**



This papaer can be improved according to the following suggestions.
1. In general, the model explanation is relatively clear, but the model design is simple, the focus of work is not prominent enough, and the sequence-attention and discourse marker are outdated.
The discourse marker was similar to Hua and Wang (2022).
Xinyu Hua and Lu Wang. 2022. Efficient argument structure extraction with transfer learning and active learning. In Findings of the Association for Computational Linguistics: ACL 2022, pages 423–437.
2. There are few comparison methods in the experimental part, which can not effectively explain the superiority of the model performance.
3. The experimental results are slightly different from the paper "Efficient Argument Structure Extraction with Transfer Learning and Active Learning", and the author needs to give corresponding explanations.
4. There are some errors, such as CACE in the legend in Figure 4, which is not mentioned in the paper.
5. In the ablation experiment, the corresponding experiment for enhancing modeling capacity (CASE+EMC) is given. The author should compare the experimental results of removing sentence-level attention module with argumentative relation regularization.


**Reasons To Accept:**

This paper proposes an Efficient Context-aware ASE model (ECASE) that fully exploits contextual information. Specifically, introduce a sequence-attention module, a distance-weighted similarity loss, and a discourse marker.

**Reasons To Reject:**




1. In general, the model explanation is relatively clear, but the model design is simple, the focus of work is not prominent enough, and the sequence-attention and discourse marker are outdated.
The discourse marker was similar to Hua and Wang (2022).
Xinyu Hua and Lu Wang. 2022. Efficient argument structure extraction with transfer learning and active learning. In Findings of the Association for Computational Linguistics: ACL 2022, pages 423–437.
2. There are few comparison methods in the experimental part, which can not effectively explain the superiority of the model performance.
3. The experimental results are slightly different from the paper "Efficient Argument Structure Extraction with Transfer Learning and Active Learning", and the author needs to give corresponding explanations.
4. There are some errors, such as CACE in the legend in Figure 4, which is not mentioned in the article.
5. In the ablation experiment, the corresponding experiment for enhancing modeling capacity (CASE+EMC) is given. The author should compare the experimental results of removing sentence-level attention module with argumentative relation regularization.


**Reproducibility:**

4: Could mostly reproduce the results, but there may be some variation because of sample variance or minor variations in their interpretation of the protocol or method.

**Reviewer Confidence:**

5: Positive that my evaluation is correct. I read the paper very carefully and I am very familiar with related work.

---

> ### Author Rebuttal · Authors · 2023-08-27
>
> We greatly appreciate the careful comments and suggestions provided by the reviewer.
>
> **Question 1**: The model design is simple, the focus of work is not prominent enough.
>
> **Answer**: In this study, we identify several issues with ASE models, including the impact of discourse markers and the distances between sentences. Based on our analysis, our goal is to propose simple yet effective solutions to these problems. Our focus is not on introducing highly novel techniques. Regarding discourse markers, they serve as crucial signals for detecting argumentative relations, a factor also emphasized in previous studies' annotations. However, we aim to reduce the models' dependence on these discourse markers and instead encourage a deeper understanding of the contextual information within the content. We believe this approach significantly diverges from previous studies. We will emphasize these differences in our forthcoming version.
>
> **Question 2**: There are few comparison methods in the experimental part.
>
> **Answer**: Our study is mainly compared with Hua et al (2022), which is shown to be SOTA in this specific task, and we also consider some other baselines, such as SVM-Linear, SVM-RBF, SEQPAIR, and SEQCON, which are the mainstream methods in this field.
>
> **Question 3**: The experimental results are slightly different from the paper "Efficient Argument Structure Extraction with Transfer Learning and Active Learning.
>
> **Answer**: We reproduce the experimental results based on the codes of Efficient Argument Structure Extraction with Transfer Learning and Active Learning (Hua et al. (2022)) released. When carrying out experiments, there are several differences in our experimental setting. First, we do not adopt data for self-supervised training of the models to adapt to different domains as Hua et al. (2022), since it increases the complexity of the model. Second, we find that when training the model in more epochs (compared with their original work), the performance is enhanced in short context windows. Thus, we report both the results of CASE they reported and those we reproduced in the manuscript.
>
> **Question 4**: There are some errors, such as CACE in the legend in Figure 4, which is not mentioned in the article.
>
> **Answer**: Sorry for the typos. It should be CASE, and we will modify this issue.
>
> **Question 5**: The author should compare the experimental results of removing sentence-level attention module with argumentative relation regularization.
>
> **Answer**: In this version, we carry out the ablation study with respect to EMC and Aug (according to the motivation of our work) to show the effectiveness of our model. We will add the results of the ablation study you mentioned in the next version.

---

### Official Review · Reviewer_d9xz · 2023-08-05

**Soundness:** 3

**Excitement:**

3: Ambivalent: It has merits (e.g., it reports state-of-the-art results, the idea is nice), but there are key weaknesses (e.g., it describes incremental work), and it can significantly benefit from another round of revision. However, I won't object to accepting it if my co-reviewers champion it.

**Paper Topic And Main Contributions:**

This paper focuses on argument structure extraction, which is formulated as a 3-way classification task. To solve this task, this paper presents several techniques including:
  1. sequence attention module on RoBERTa (concat the [CLS] tokens of all sentences in the context window).
  2. distance-weighted similarity loss.
  3. data augmentation with randomly masked discourse markers and randomly masked random sentences.

**Reasons To Accept:**

- The techniques are presented clearly and contribute to the performance.
- The presented method ECASE outperforms the baselines in most of the settings.

**Reasons To Reject:**

- The results of the ablation studies are not quite clear. The authors could at least plot more horizontal lines or display the numbers. In the current plot, the improvement of ECASE over CASE+EMC seems marginal on AbstRCT, AMPERE, and Essays.
- All the techniques seem marginal and irrelevant. This paper can have a better story if the techniques of concatenating [CLS] tokens of in-context sentences and data augmentation can be more connected.

**Reproducibility:**

3: Could reproduce the results with some difficulty. The settings of parameters are underspecified or subjectively determined; the training/evaluation data are not widely available.

**Reviewer Confidence:**

3: Pretty sure, but there's a chance I missed something. Although I have a good feel for this area in general, I did not carefully check the paper's details, e.g., the math, experimental design, or novelty.

---

> ### Author Rebuttal · Authors · 2023-08-27
>
> We greatly appreciate the careful comments and suggestions provided by the reviewer.
>
> **Question 1**: The results of the ablation studies are not quite clear.
>
> **Answer**: We will improve the visualization of the results. Given the significant differences in language expressions across these domains, the effects of the EMC and Aug modules also vary. But generally, ECASE achieves stronger performance compared with CASE-EMC and CASE-Aug, and a significant improvement compared with the baseline CASE.
>
> **Question 2**: All the techniques seem marginal and irrelevant.
>
> **Answer**: In this study, our primary focus is on the efficient way to utilize the contextual information inherent in the data. We approach this problem from various perspectives, including data analysis and modeling capacity, the latter of which encompasses both model structure and training objectives. For modeling capacity, we introduce a sequence-attention module and distance-weighted similarity loss to aggregate contextual information and argumentative information. For data augmentation, we augment the training data by randomly masking discourse markers and sentences, which reduces the model's reliance on specific words or less informative sentences.

---

### Official Review · Reviewer_LkxQ · 2023-08-12

**Soundness:** 4

**Excitement:**

4: Strong: This paper deepens the understanding of some phenomenon or lowers the barriers to an existing research direction.

**Paper Topic And Main Contributions:**

The paper proposes an efficient context-aware framework for argument structure extraction (ASE). The method, ECASE, aims to enhance the model capacity by learning the context more efficiently than context windows, while taking advantage of a data augmentation that helps the model learn from more informative sentences. The model capacity is enhanced via an attention module. The data augmentation is implemented through randomly masking discourse markers and sentences. The experiments show that ECASE outperforms context-aware baselines on a suite of benchmark datasets, and the ablation study confirms the effectiveness of both model capacity enhancement and data augmentation towards the success of ECASE.

**Questions For The Authors:**

- (A)  It is quite puzzling that both CASE+Aug and CASE+EMC perform better than CASE on AbstRCT, but combining Aug with EMC slightly hurts the performance. Is it the masking from Aug that is making EMC less effective, or is it the additional loss term from EMC that is making training on the augmented data more difficult? Which one of Aug and EMC is reducing the gain from the other?

**Reasons To Accept:**

- The core idea behind the development of this method is clear and well-motivated to improve context-aware ASE such that the model learns from informative but distant context without being negatively impacted by the length of context.

- The method is well-designed and clearly explained, and it can accommodate custom changes or additions to its various components for future developments.

- The authors have appropriately introduced a weighted regularization term in the training loss to take advantage of similarities between representations of propositions with argumentative relation.

- The experiments are thorough with sufficient comparison with benchmark models some of which are specifically designed for context-aware ASE.

- The experiments show the success of ECASE in achieving its intended goals. In particular, the ablation study shows that both model capacity enhancement and data augmentation are improving the performance.

**Reasons To Reject:**

- While not a weakness of the method itself, why ECASE seems to perform slightly worse than CASE+Aug and CASE+EMC on the AbstRCT dataset (Figure 4) is not addressed. Perhaps a discussion of specific features of this dataset that the authors think could cause this could help.

**Reproducibility:**

4: Could mostly reproduce the results, but there may be some variation because of sample variance or minor variations in their interpretation of the protocol or method.

**Reviewer Confidence:**

3: Pretty sure, but there's a chance I missed something. Although I have a good feel for this area in general, I did not carefully check the paper's details, e.g., the math, experimental design, or novelty.

**Typos Grammar Style And Presentation Improvements:**

- It seems like the first legend in Figure 4 should be “CASE” (instead of “CACE”).

- Please spell out "PLM" before using the abbreviated form (line 158).

- Although clear to familiar readers, it would be stylistically better to briefly mention what $\mathcal{L}_{cls}$ (line 188) is in the text. It is only specified in Figure 3.

- Experiments with different head to tail distances are reported in Appendix E, but since the introduction specifically motivates the research to learn efficiently from longer context, this deserves at least a reference to the appendix in the main text.

---

> ### Author Rebuttal · Authors · 2023-08-27
>
> We greatly appreciate the careful comments and suggestions provided by the reviewer.
>
> **Question**: Why ECASE seems to perform slightly worse than CASE+Aug and CASE+EMC on the AbstRCT dataset.
>
> **Answer**: For the reviewer's concern, given the significant differences in language expressions across these domains, the effects of the EMC and Aug modules also vary. When these modules are combined, the information complexity increases, potentially necessitating additional steps to achieve optimal performance. We have kept the training hyper-parameters consistent across all datasets to maintain fairness. Our findings indicate that extending the training duration improves the performance of ECASE in Abst_RCT compared to CASE-EMC and CASE-Aug. We will incorporate this observation into the description in our forthcoming version.
>
> We acknowledge the typographical errors and will rectify them in the next version. We will also modify the descriptions as you suggest.

---

### Meta-Review · Area_Chair_oMFD · 2023-09-27

**Recommendation:** 3

**Metareview:**

This paper introduces a context-aware framework for argument structure extraction (ASE). The method, ECASE, consists of a sequence attention module, a distance-weighted similarity loss, and a discourse marker masking technique. The paper evaluates ECASE on five benchmark datasets and shows that it outperforms context-aware baselines on most of them.

The reviewers have mixed opinions about the paper. Reviewer 1 is very positive and praises the paper for its clear motivation, well-designed method, and thorough experiments ablation study. Reviewer 2 is ambivalent and gives two reasons to accept the paper (clear presentation and superior performance) and two reasons to reject the paper (unclear ablation results and marginal techniques). The reviewer was not convinced by the authors' response. Reviewer 3 is borderline and gives only one reason to accept the paper (exploiting contextual information) and several reasons to reject the paper (outdated model design, lack of comparison methods, inconsistency with a previous paper, and missing ablation experiment). This reviewer didn't change their opinion after reading the authors' response to these concerns.

Based on these reviews and other comments, I think the paper has some merits in proposing an efficient context-aware ASE model that achieves state-of-the-art results on some datasets, but it also has some weaknesses in its comparison with older work in terms of model design, ablation analysis, and presentation quality. The authors should address the reviewers’ comments and questions in their revision and clarify their contributions and limitations.

---

### Decision · Program_Chairs · 2023-10-07

**Decision:**

Accept-Findings

**Comment:**

This paper introduces a context-aware framework for argument structure extraction (ASE). The method, ECASE, consists of a sequence attention module, a distance-weighted similarity loss, and a discourse marker masking technique. The paper evaluates ECASE on five benchmark datasets and shows that it outperforms context-aware baselines on most of them.

The reviewers have mixed opinions about the paper. Reviewer 1 is very positive and praises the paper for its clear motivation, well-designed method, and thorough experiments ablation study. Reviewer 2 is ambivalent and gives two reasons to accept the paper (clear presentation and superior performance) and two reasons to reject the paper (unclear ablation results and marginal techniques). The reviewer was not convinced by the authors' response. Reviewer 3 is borderline and gives only one reason to accept the paper (exploiting contextual information) and several reasons to reject the paper (outdated model design, lack of comparison methods, inconsistency with a previous paper, and missing ablation experiment). This reviewer didn't change their opinion after reading the authors' response to these concerns.

Based on these reviews and other comments, I think the paper has some merits in proposing an efficient context-aware ASE model that achieves state-of-the-art results on some datasets, but it also has some weaknesses in its comparison with older work in terms of model design, ablation analysis, and presentation quality. The authors should address the reviewers’ comments and questions in their revision and clarify their contributions and limitations.